# Evaluating the Investment Efficiency of China's Provincial Power Grid Enterprises under New Electricity Market Reform: Empirical Evidence Based on Three-Stage DEA Model

**Jingqi Sun [1,2], Nuermaimaiti Ruze [1,2,*], Jianjun Zhang [1,2], Haoran Zhao [1,2] and Boyang Shen [3]**

[1] School of Economics and Management, North China Electric Power University, Beijing 102206, China; sjq@ncepu.edu.cn (J.S.); 1182206100@ncepu.edu.cn (J.Z.); 1162106030@ncepu.edu.cn (H.Z.)

[2] Beijing Key Laboratory of New Energy and Low-Carbon Development, North China Electric Power University, Changping District, Beijing 102206, China

[3] Electrical Engineering Division, Department of Engineering, University of Cambridge, Cambridge CB3 0FA, UK; bs506@cam.ac.uk

* Correspondence: nruze@ncepu.edu.cn

**Abstract:** The new round of electricity market reform in 2015 completely changed the profit pattern of power grid enterprises (PGEs) in China, and directly affected their investment plans. Under the new electricity market reform (NEMR), the government regulatory authority made higher requirements for the investment efficiency of PGEs, and the investment effectiveness hence became the core criterion for investment plans. Therefore, the PGEs are now attaching great importance to the investment efficiency. According to their geographical differences, this paper divides the Chinese provincial PGEs into three groups, namely eastern, central and western region enterprises. Based on the NEMR, we developed an evaluation system of investment efficiency for the above-mentioned enterprises. Moreover, this paper selects GDP per capita, electricity consumption in industry, and electrification rate as external environment variables, and conducts an empirical research on the investment efficiency of 31 provincial PGEs in China in 2017. The analysis reveals that three external environment variables have considerable impacts on the investment efficiency. Though the increase of GDP per capita and electricity consumption in industry are not conducive to improving investment efficiency, the advancement of electrification plays a positive role in its improvement. And from the real efficiency results, Tianjin, Liaoning, Jiangsu, and Fujian have relatively higher investment efficiency, while Henan, Shandong, and Shanghai exhibit lower investment efficiency. By comparing the investment efficiency of PGEs in the first and third stage, conclusions can be drawn that in the first stage the investment efficiency of PGEs was overestimated, and the inefficient investments prevailed some provincial PGEs, which caused by low scale efficiency.

**Keywords:** new electricity market reform; government regulation; three-stage DEA model; investment efficiency; external environment variables

## 1. Introduction

In the modern society, the development of electric power industry is a critical index of a country's economy. The PGEs, the bridge between electricity generation and utilization, are an indispensable component of electric power industry. Therefore, the investment of PGEs plays a vital role in the development of a nation's economy. In March 2015, the State Council of Central Committee of CPC issued the "Opinions on Further Deepening the Structural Reform of Electric Power" (C [2015] No. 9)



(referred to as Document No. 9) [1] to resolve the striking but fundamental problems which limited the scientific development of the industry, and also to promote its sound and rapid development. The "Opinions" requires a strengthened overall planning of the electric power industry, especially of the PGEs, and an enhanced supervision and regulation system for the fair access of power grid, grid investment behavior, and cost/investment efficiency. The structural reform of electric power generates twofold influences. On the one hand, as the core effect, the new round of structural reform of electric power with the reform of transmission and distribution price has completely changed the profit pattern of PGEs. Power companies and users therefore are able to set price through direct consultation with power plants. Consequently, the PGEs are merely in the position of providing power transmission and distribution services. Meanwhile, the profit of PGEs will mainly come from the relatively fixed cost of power flow [2]. On the other hand, the relevant government authorities are obliged to conduct comprehensive supervision and regulation towards the investment of PGEs in terms of the reliability of power supply, power supply capacity, and economic, social, and environmental benefits, etc. Overall, the NEMR has profoundly modified the production model of PGEs in all aspects, including network planning, investment, construction, operation, and maintenance.

Following the NEMR, the effective asset of PGEs becomes the essential factor affecting the power transmission and distribution price. Investment effectiveness will become the core criterion for grid enterprise investment. The National Development and Reform Commission (NDRC) has promulgated a series of regulations to normalize the reform of transmission and distribution prices, but also impose stricter requirements on the supervision of PGEs investment. The "Regulations on the Pricing of Transmission and Distribution Price of Provincial Power Grids (Trial)" attaches more importance to enhance the supervision of PGEs' investment from its source. The "Notice on Expanding the Scope of the Pilot Program of Reforming the Transmission and Distribution Price" clearly states that it is necessary to explore and establish a post-investment evaluation mechanism for PGEs, and their unreasonable and ineffective investments and costs will not be included in the transmission and distribution price, which means PGEs nowadays are facing even stricter supervision and regulation. Meanwhile, the PGEs need to improve their input and output efficiency and avoid prevailing problems such as pursuing the quantity and scale of investment and equipment replacement in blind, investing projects beyond ability, and producing without efficiency, which can undoubtedly limit their ability to achieve the strategic objectives in the transforming period, and adapt to the increasingly disagreeable business environment and regulatory requirements. It is urgent for PGEs to develop a scientific quantitative evaluation model to calculate the investment efficiency including various parameters such as technology, economy, and society. On that account, the more objective and comprehensive assessment will enable the government to make more reasonable decisions.

At present, many scholars have studied the investment decision-making process, investment capacity and the accurate investment of PGEs, but the investment efficiency of these enterprises under the NEMR is still a less touched topic. Many industries have already analyzed their investment efficiency, including agriculture, construction, finance, and social environment. However, the efficiency of investment in the electric power industry, especially in PGEs, has not received its deserved attention. He et al. [3] studied the impact of transmission and distribution price policy on the development of China's power market and PGEs. Based on the theory of system dynamics, they put forward policy suggestions for the investment plans and investment capabilities of PGEs. Dunnan, Liu and Erfenga, Xu et al. [4] analyzed the effect of transmission and distribution price reform on the investment scale of power grid, and they proposed an investment demand estimation model for power grid and an investment capacity measurement model for PGEs.

Various methods can be used to measure investment efficiency, such as the econometric methods and data envelopment analysis (DEA). Based on many rigorous assumptions, econometric methods have high requirements for time series. The electric power industry is a particularly complex system with countless uncertainties, which is unlikely to meet the basic requirements of econometric model. DEA method is a practical linear programming method, and is capable of establishing an investment

efficiency evaluation system for the electric power industry under special circumstances, to more accurately measure the efficiency of investment in PGEs. Since the DEA model was introduced by Charnes et al. [5] in 1978, both the development of the theory and its application to actual cases have achieved remarkable advancement. In general, the DEA model is highly applicable to the complex power grid company systems, which have multiple input and output indicators.

With the development of society, science, and technology, scholars have found that traditional DEA model is liable to some errors and excludes the impact of external uncontrollable variables on efficiency. After gradual improvements, the traditional DEA model upgraded to two-stage DEA and three-stage DEA model. Though many studies used three-stage DEA model to analyze the electric power industry, they merely touched limited areas, such as power grid operation, economic efficiency, and social and environmental efficiency. This paper combines the traditional DEA model with three-stage DEA model, and excluded factors such as external environment variables, the effect of random errors and the influence of inefficient management on the investment efficiency of Chinese provincial PGEs through stochastic frontier analysis (SFA) model.

There is a big gap between China's eastern, central, and western economic levels and education levels. China's provincial PGEs are considered in three regions, thus eliminating the impact of regional differences on the investment efficiency of PGEs. Therefore, this paper divides 31 Chinese provincial PGEs into three groups, namely eastern, central and western region enterprises according to their geographical differences, and develops an investment efficiency evaluation system for PGEs to conduct an empirical research of the input and output efficiency of these enterprises in 2017.

The rest of this paper is organized as below: Section 2 is the literature review of existing researches on traditional DEA and three-stage DEA. The application of three-stage DEA model is elaborated in Section 3. Section 4 explicitly explains the investment efficiency index system for PGEs in this paper, which is set under the NEMR and the transmission and distribution price reform. Section 5 presents the empirical research results of the investment efficiency evaluation of 31 provincial PGEs. Section 6 summarizes the whole paper and discusses the results.

## 2. Literature Review

Data envelopment analysis (DEA) is a mature method to evaluate the unit efficiency in many fields, and it has the possibility to be a good evaluation tool for analyzing the efficiency of various industries [6–11]. Jingmin, Wang and Yufang, and Shi et al. [12] applied the Bootstrap-DEA model to evaluate the energy efficiency of the industrial sector in Beijing. It shows that the overall industrial energy efficiency is high and still rising, but the result of different industries varies considerably. Li and Jiang et al. [13] analyzed the agricultural efficiency in Chinese provinces based on DEA model, measured the overall technical efficiency of agriculture in China, and calculated the energy saving potential of each province in accordance with their efficiency. Liu et al. [14] used energy value method and DEA model to evaluate the ecological efficiency of China's coal recycling economy system from 2006 to 2015, and concluded that the eco-efficiency is positively correlated with economic benefit, and recycling industry will be signally conducive to energy-saving and emission-reducing. George Halkos et al. [15] resorted to DEA model to study the environmental efficiency of pollutant emissions in 28 EU member states, suggesting that Germany, United Kingdom and Ireland have the highest environmental efficiency.

DEA model has been used to explore electric power industry for a relatively long period of time, and there are plenty of documented literatures on this topic. Hossein et al. [16] proposed an optimization algorithm of wind farm location based on the fuzzy cross-efficiency DEA model to accurately calculate the efficiency of the location. Rajiv Banker et al. [17] examined the application of DEA in incentive regulation of power distribution enterprises, and found that the frontier of production cost estimated by DEA model was important in the setting of electricity price. Mullarkey et al. [18] applied the DEA framework to calculate the technical efficiency of the Irish distribution network and the promoted efficiency through the reorganization and consolidation of the Irish distribution

network. Combining DEA and DEA-discriminant analysis, Toshiyuki et al. [19] studied the operating performance of PGEs after the liberalization reform of the Japanese power industry in 2005. Based on DEA model, Mika Goto et al. [20] surveyed the effects of market reform and Fukushima nuclear power plant accident on the operating efficiency and environmental efficiency of nine Japanese power companies through the data collected during 2003 to 2015. The literatures above suggest that though at present DEA is a mature and modern tool for efficiency calculation, it still cannot avoid some shortcomings. The traditional DEA model does not consider the effects of some uncertain factors, such as external environment variables and statistical errors. Hence, this paper adopts a more rigorous method, i.e., three-stage DEA model.

Compared with traditional DEA model, three-stage DEA model has the greatest advantage in analyzing the influence of uncontrollable factors such as external environment variables, random errors and management inefficiency. Zhao Haoran et al. [21] availed three-stage DEA to accurately eliminate external factors and studied the energy efficiency of each province in China in 2008–2016. Zhang et al. [22] used three-stage DEA model to measure the technical efficiency of the regional construction industry affected by environmental regulation policies. The study showed that environmental regulation was particularly efficient for the construction industry in China. Hong Kuan et al. [23] explored the innovation efficiency of China's semiconductor industry based on the generalized three-stage DEA model, and concluded that input redundancy was common in the manufacturing and equipment of the whole industry. Zhao Changhong et al. [24] measured the total factor energy efficiency (TFEE) of 35 "Belt and Road" initiative countries in 2015 based on three-stage DEA model. Xie and Duan et al. [25] conducted a three-stage DEA analysis of the environmental efficiency of 37 industry sectors in China. The analysis showed that uncontrollable variables had a striking impact on environment efficiency. Zeng and Hu et al. [26] conducted an empirical study on the investment efficiency of Chinese cultural industry in 2011 using three-stage DEA model, revealing that the scale of enterprises was a critical factor restricting the development of enterprises in the cultural industry. Ebrahimnejad et al. [27] applied three-stage DEA into the measurement of banking industry performance. The validity of the method, together with the applicability of the method in banking performance evaluation was verified by an example analysis. I.M. García-Sánchez [28] studied the efficiency and effectiveness of the Spanish professional football league through three-stage DEA, whose results indicated that the technical inefficiency of defense was greater than the offense, which was caused by poor player ability management and team size. Jui-Kou Shang [29] used three-stage data envelopment analysis method to analyze the impact of e-commerce on hotel performance, suggesting that there was no significant difference in efficiency due to their different application conditions of e-commerce.

Thus, three-stage DEA model has already been applied in various industries. The three-stage DEA model is on the basis of traditional DEA model, and can eliminate uncontrollable external environment variables in each industry and the influence of random error terms on decision-making efficiency, which reflects the real efficiency of decision-making units. Chen and Li et al. [30] evaluated the low carbon benefits of smart power grid systems. They introduced a three-stage DEA model and established a low-carbon benefit evaluation system in accordance with the characteristics of the national smart power grid, and provided guidance for the scientific layout and development of the smart power grid. Under new round of electricity market reform, Zhao and Guo et al. [31], for the first time, used three-stage DEA model to accurately evaluate the actual operating efficiency of 26 provincial PGEs in China, taking into account the effect of external environment variables on the operation efficiency of the provincial power grid.

As discussed above, three-stage DEA model is widely used in scientific research on efficiency. Its analysis is reliable and can reflect the real efficiency of the decision-making units (DMUs). Therefore, three-stage DEA model is more practical in calculating the efficiency.

## 3. Research Methods

DEA is an efficiency evaluation methodology proposed by the famous American operations researchers Charnes, Cooper and Rhodes in 1978 [5]. It utilizes the principle of mathematical programming to obtain efficiency based on multiple sets of input and output data, and the total efficiency value is the product of the configuration efficiency and the technical efficiency. In 1984, Banker, Charnes, and Cooper all proposed a more rigorous correction model called the BCC (Banker Charnes and Cooper) model [32]. The BCC model changes the fixed-scale compensation of the CCR (Charnes Cooper and Rhodes) model to variable-scale compensation [32], so that the comprehensive efficiency in the BCC model is divided into the pure technical efficiency and scale efficiency, which is, comprehensive efficiency = scale efficiency (SE) × pure technical efficiency (PTE). However, the disadvantages of the two models above are that they do not consider the external environmental factors and the impact of random errors in the overall efficiency.

In the two papers published by Fried (1999, 2002), a new three-stage DEA model was proposed based on the previous research [33,34]. The three-stage DEA model analysis can overcome the limitations of traditional DEA analysis to ignore the influence of external environment variables. The advantage of the three-stage DEA model over the traditional DEA model is that it can eliminate the influence of environmental factors, statistical noise, and management inefficiencies on the efficiency measurement of decision-making unit (DMU). The stage of the three-stage DEA is shown in Figure 1. In the first stage, the input-oriented BCC model is used to calculate the overall efficiency under the influence of environmental factors and statistical noise, but also to calculate the slack variables value of the original input. In the second stage, based on the Stochastic Frontier Analysis (SFA) method, the influence of stochastic noise and external environment values on the efficiency estimation is removed. Meanwhile, the slack variables are taken as the explained variable, and the external environment variables are taken as the explanatory variable. In the last stage, original input variables are replaced by the adjusted input variables, and the BCC model is used, so as to obtain a more accurate value of efficiency.

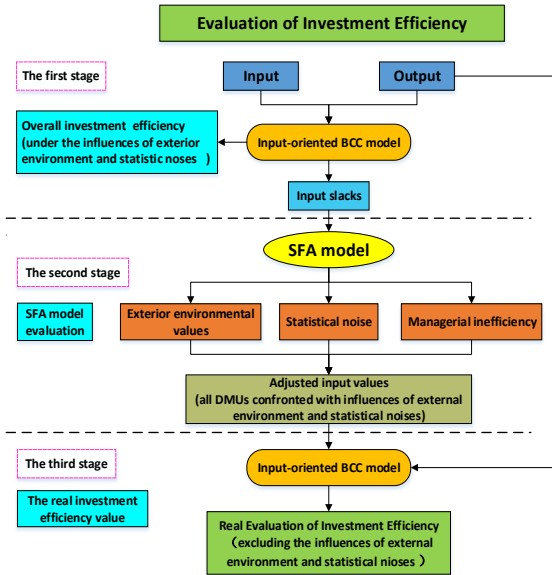

**Figure 1.** Research methods for investment efficiency of power grid enterprises (PGEs). SFA: stochastic frontier analysis.

### 3.1. Stage 1: The Traditional DEA Model Analysis of the Original Input and Output Values

At this stage, the initial input–output data of 31 DMUs are used, and the BCC correction model with variable returns of scale is used for DEA analysis. Thus, the overall efficiency values of the 31 DMUs are calculated separately. In the BCC model:

$$Min\ \theta$$
$$s.t.$$
$$\sum_{j=1}^{n} \lambda_j y_{rj} \geq y_{rj0}$$

$$\sum_{j=1}^{n} \lambda_j x_{ij} \leq x_{ij0}$$

$$\sum_{j}^{n} \lambda_j = 1,\ \lambda_j \geq 0 \tag{1}$$

$$j = 1, 2, \ldots, n;$$
$$i = 1, 2, \ldots, m;$$
$$r = 1, 2, \ldots, s;$$

where $\theta$ demonstrates the comprehensive operational efficiency value of each DMU, $x_{ij}$ and $y_{rj}$ are the $i$-th input and $r$-th output of the $j$-th DMU, m, s, and n represent the amount of input variables, output variables, and DMUs, and $\lambda_j$ implies a j dimensional weight vector of the DMU $j$.

### 3.2. Stage 2: SFA Regression Model

After the first stage, the overall provincial investment efficiencies for DMUs can be acquired. Meanwhile, the sequences of all input variables can be calculated. The input slacks of all DMUs are influenced by the external environment parameters, managerial inefficiency, and statistical noises. At this stage, they can be eliminated by the SFA approach. Finally, the adjusted optimal input value is obtained.

According to the research method of Fried et al., the calculation method of the input slack variable value is as shown in Equation (2):

$$S_{in} = x_{ni} - \lambda X_n \geq 0 \tag{2}$$

where $S_{in}$ implies the input slack variable values, $x_{ni}$ represents the original value of the input variables, $\lambda X_n$ illustrates the ideal optimal input values.

The input-oriented SFA regression model is listed in Equation (3):

$$S_{ni} = f(Z_i; \beta_n) + \varepsilon \tag{3}$$

$$\varepsilon = v_{ni} + u_{ni} \tag{4}$$

where $Z_i$ is the exterior environmental values, $\beta_n$ implies The coefficients of the environmental variables, $u_{ni}$ represents the administration inefficiency term, subject to a normal distribution truncated at zero, where $u \sim N^+(0, \sigma_u^2)$, $v_{ni}$ illustrates the statistical noises, where $v \sim N(0, \sigma_v^2)$, $v_{ni} + u_{ni}$ implies mixed error term.

Setting $\gamma = \sigma_u^2 / \sigma_v^2 + \sigma_u^2$, and the value of $\gamma$ close to 1 demonstrates that the impacts of management inefficiency dominate the investment inefficiencies of DMUs, and thus the SFA approach could be utilized for estimation. By contrast, the value of $\gamma$ near to zero indicates the investment inefficiency of DMUs are mainly derived from statistical noises, and thus the ordinary least squares (OLS) approach can be applied for measurement. Therefore, the value of $\gamma$ is utilized to identify the feasibility and applicability of the SFA approach for regression.

To adjust the input of each DMUs, mainly refers to the decomposition method of Jondrow [35], and the conditional expectation value of managerial inefficiency component is calculated by:

$$E(u/\varepsilon) = \sigma_* \left[ \frac{\varnothing\left(\lambda\frac{\varepsilon}{\sigma}\right)}{\Phi\left(\frac{\lambda\varepsilon}{\sigma}\right)} + \frac{\lambda\varepsilon}{\sigma} \right] \tag{5}$$

where $\sigma_* = \frac{\sigma_u\sigma_v}{\sigma}$, $\sigma = \sqrt{\sigma_u^2 + \sigma_v^2}$, $\lambda = \sigma_u/\sigma_v$, respectively.

Afterwards, based on the SFA approach for estimating the values of $\beta_n$, the conditional expectation values of $v_{ni}$ is computed by:

$$E[v_{ni}|v_{ni} + u_{ni}] = S_{ni} - f(Z_i; \beta_n) - E[u_{ni}|v_{ni} + u_{ni}] \tag{6}$$

Then, the adjusted input values $X_{ni}^A$ are calculated by:

$$X_{ni}^A = X_{ni} + \left[ max\left( f\left(Z_i + \hat{\beta}_n\right) - f\left(Z_i + \hat{\beta}_n\right) \right) \right] + [max(v_{ni}) - v_{ni}] \tag{7}$$

The component $[max(v_{ni}) - v_{ni}]$ in the Equation (7) implies the adjustment dispatching all the DMUs in the least desirable environment confronted by all DMUs. The term $\left[ max\left( f\left(Z_i + \hat{\beta}_n\right) - f\left(Z_i + \hat{\beta}_n\right) \right) \right]$ adjusts DMUs to a common condition of nature called the unluckiest condition confronted by all the samples. The adjustment above dispatches all of the DMUs in the same exotic environment.

### 3.3. Stage 3: Adjusted Efficiency Value

After adjusting the input slacks in the second phase, the investment value $x_{ni}^a$ is as new variables replacing the original input value $x_{in}$, and then the BCC model is used again. Finally, the real efficiency value is obtained after excluding exterior environmental variables and statistical noises. The third phase is similar to the first phase, the details for the third stage can refer to [34].

## 4. Index System

### 4.1. Selection of Input Variables

The selection of index variables is the most important part of the three-stage DEA model. According to the principle of production function, the development of power grid depends on the input of labor, finance, and material. And this paper selects output indexes from economic, environmental and social benefits. Considering the availability of raw data, this paper collects original input and output data in 2017 of 31 provincial PGEs from 22 provinces, five autonomous regions (Xinjiang, Ningxia, Guangxi, Inner Mongolia, Tibet) and four municipalities (Beijing, Shanghai, Chongqing, Tianjin). Since the index data of Taiwan, Hong Kong, and Macao is not available, it is not included in the scope of this study.

According to the investment regulatory assessment criteria issued by the government, under the new round of electric power reform, whether the new investment of PGEs can form effective assets is a key factor in investment decision-making. Therefore, when PGEs form investment plans, besides considering business development, the formation of effective grid investment assets will be further emphasized. This paper fully considers the special background of PGEs under the reform of transmission and distribution price, and selects the investment of grid infrastructure, the size of employees and the variable capacity of 35kV and above as the input variables of three-stage DEA model.

### 4.2. Selection of Output Variables

Under the new electricity power reform, the output of PGEs is mainly reflected in power supply reliability, quality service, customer satisfaction, environmental benefits, and economic benefits. This paper selects marketized trading electricity, the power supply reliability, electric

energy replacement, and integrated line loss rate as output indexes to conduct empirical research on the investment efficiency of PGEs.

Marketized trading electricity reflected the opening up of PGEs under the electric power system reform. It is the core content of breaking monopoly and introducing competition into the industry. Meanwhile, it is an important index, which reflects the degree of the promotion of PGEs' system reform and the effectiveness of electric power market reform. The power supply reliability rate = (the user effective power supply time/the statistical period time) × 100%. The reliability of power supply reflects the power supply capacity and power supply reliability of the power grid system, and is one of the most important indexes that affect customer satisfaction. Electric energy replacement occurs in the final energy consumption. It means using electric energy to replace the traditional energy consumption mode of loose coal and fuel oil burning. By actively advocating the new energy consumption mode of "replacing coal by electricity, replacing oil by electricity, and get electricity from afar" will continuously increase the proportion of electrical energy in the final energy consumption. Electric energy replacement has the advantages of clean, safe, and convenient, which has been promoting the energy consumption revolution. The replacement of electricity by electric energy is one of the important indexes reflecting the environmental benefits of PGEs under the new electricity reform. The integrated line loss rate is the ratio of the power consumed and lost by the line in the power supply production process to the power supply. The rate is equal to the difference between the amount of power supplied and the amount of electricity sold that accounts for the proportion of power supplied. The integrated line loss rate is a comprehensive technical and economic index reflecting the level of power management and technical management, and the power supply capacity of the power grid.

### 4.3. The Selection of External Environment Variables

It is not only the internal input and output of the PGEs that affects the investment efficiency, but also the external environment factors do so. The selection of external environment variables is critical to the setting of three-stage DEA model. The investment efficiency of PGEs is influenced by many external environment factors. In addition to the effect of transmission and distribution price reform, they also include local socio-economic development and industrial development. These factors will directly affect the power supply and power sales of PGEs, and thus affect their investment efficiency. This paper selects indexes such as GDP per capita, the proportion of industrial electricity consumption (TPIEC), and electrification rate as external environment variables.

Although China's three regions eliminate the impact of geographical differences on the investment efficiency of PGEs, but it cannot eliminate the impact of local economic level on the investment efficiency of PGEs. The development of PGEs is highly correlated with the regional economic developing level. Meanwhile, the living standard of local people is also determined by the economic development level, which can affect the electricity consumption amount, thereby influencing the electricity sales amount of PGEs. While considering this, the GDP per capita (yuan per person) representing the regional economic development level is selected as an exotic environmental variable affecting operational efficiency of provincial PGEs. The proportion of industrial electricity consumption (%) is the proportion of industrial electricity consumption in the whole society, and it reflects the electricity consumption of local industrial structure. At present, the indexes measuring the electrification rate (%) of a country or a region mainly are electricity consumption per capita, power generation energy accounted for the proportion of primary energy and electricity accounted for the proportion of final energy consumption. Among them, electricity consumption per capita of each country varies greatly due to the differences in geographical location, resource environment, economy structure and population. The proportion of power generation in primary energy and the proportion of electrical energy in final energy consumption reflect the position of electrical energy in the entire energy system, and are internationally used electrification indexes. At the same time, the proportion of electrical energy in final energy consumption is an important index reflecting the power economic activities in a region [36,37].

Hence this paper selects it to represent the electrification rate of various provinces in China, and we set an evaluation index system for investment efficiency of provincial PGEs, as shown in Figure 2.

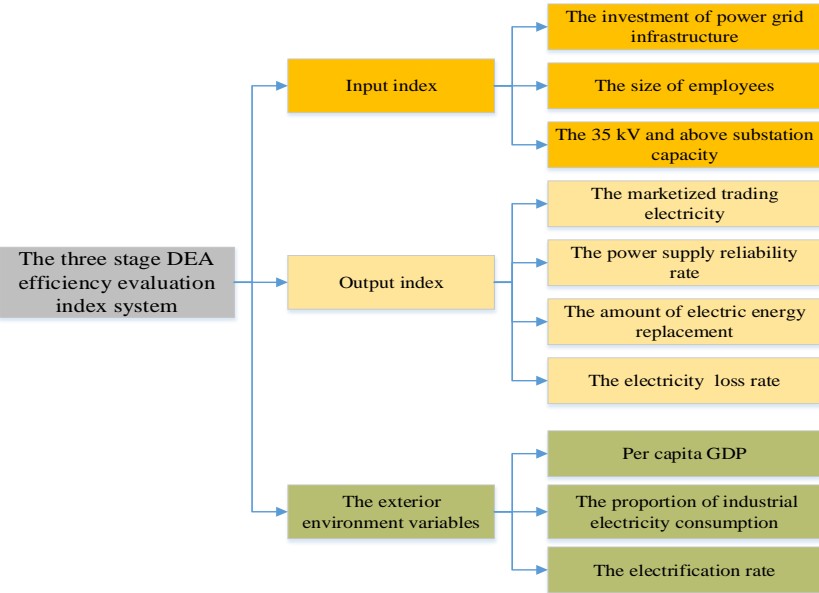

**Figure 2.** The index system for investment efficiency of PGEs. DEA: data envelopment analysis.

### 4.4. Data Sources and Processing

The data of input and output indexes and the indexes of social variables is mainly collected from the National Bureau of Statistics of China [38], the Bureau of Statistics of each province, the official website of each provincial power grid enterprise, China Electric Power Yearbook, China Statistical Yearbook 2018 [39], etc. Data related to fixed assets investment, personnel size, and line length are collected from the 2017 Corporate Social Responsibility Annual Report of each provincial power grid enterprise and China Electric Power Yearbook (shown in Table 1). Data of marketized trading electricity, power supply reliability rate, electric energy replacement, line loss rate is collected from China Electric Power Yearbook. Data such as GDP per capita, industrial electricity consumption, and electrification rate is collected from the official website of the National Bureau of Statistics of China and the Bureau of Statistics of each province, as well as public publications such as the China Statistical Yearbook 2018.

**Table 1.** Descriptive statistics of selected variables for 31 provincial PGEs.

| Variables | Minimum | Maximum | Mean |
|---|---|---|---|
| The investment of grid infrastructure ($10^8$ Yuan) | 35 | 562.03 | 159.95 |
| The size of employees (persons) | 4809 | 149,501 | 49,936 |
| The 35 kV and above substation capacity (10 MVA) | 938.6 | 47,500 | 15,703.35 |
| The marketized trading electricity ($10^8$ KWh) | 10.59 | 1618 | 508.35 |
| The electric energy replacement ($10^8$ KWh) | 2.29 | 130.25 | 44.84 |
| The integrated line loss (%) | 3.49 | 13.74 | 6.5 |
| The power supply reliability (%) | 99.36 | 99.991 | 99.85 |

The integrated electric loss rate in the output variable is as small as possible, and contrary to other variables. Therefore, in order to consistent the output variable, the overall electric loss rate is reciprocally processed.

## 5. Empirical Research

### 5.1. The First Stage: Analysis of Integrated Investment Efficiency

In this stage, without considering the impact of external environment variables, using the modified input-oriented BCC model can count out the integrated investment efficiency of 31 Chinese provincial PGEs (the efficiency of the first stage and the third stage will be put in Table 2). The integrated investment efficiency of each DMUs reflects the input efficiency of the PGEs. From Table 2, it can be seen that the average integrated efficiency of these provincial PGEs is 0.844, the average pure technical efficiency is 0.934, and the average scale efficiency is 0.895. These results show that although the investment efficiency of provincial PGEs is relatively high, there is still some room for improvement. From the perspective of regional differences, the average efficiency of provincial PGEs in the eastern, central and western region is 0.816, 0.80, and 0.906 respectively. The provincial PGEs in the western region have the highest investment efficiency under the NEMR, followed by the eastern part. The central region has the lowest efficiency, which indicates that the investment plans and management of provincial PGEs in the central region need improvements.

**Table 2.** The SFA estimation results.

| The Slack Variables | The Investment of Grid Infrastructure | | The 35 kV and Above Substation Capacity | | The Size of Employees | |
|---|---|---|---|---|---|---|
| | Coefficient | T Test Values | Coefficient | T Test Values | Coefficient | T Test Values |
| Constant term | −99.77 | −222.09 * | −8058.92 | −1164 * | −33,562.23 | −33,562 * |
| GDP per capita | $6.97 \times 10^{-4}$ | 41.25 * | 0.0457 | 121.87 * | 0.1118 | 22.995 * |
| TPIEC | 81.05 | 107.31 * | 5705.77 | 444.01 * | 55,258.53 | 55,258 * |
| Electrification level | −83.14 | −108.82 * | −2537.93 | −220.5 * | −47,297.02 | −47297 * |
| $\sigma^2$ | 10,540.29 | 10,540.36 | $3.71 \times 10^7$ | $3.7 \times 10^7$ | $1.51 \times 10^9$ | $1.51 \times 10^9$ |
| $\gamma$ | 0.999999 | 565,551.5 | 0.999999 | 1,790,472 | 0.999999 | $1.49 \times 10^6$ |
| Log likelihood | −164.7 | | −292.185 | | −348.13 | |
| LR test | 17.52 * | | 15.14 * | | 18.22 * | |

Note: * indicates 1% significance level. TPIEC: the proportion of industrial electricity consumption.

As can be seen from Table 2, provincial PGEs of Beijing, Tianjin, Liaoning, and Jiangsu (integrated efficiency is 1) are better at the investment with promising integrated investment efficiency. The investment efficiencies of provincial PGEs in three economically developed regions such as Shandong, Shanghai, and Henan are less than 50%, indicating that these enterprises have inefficient investments and waste resources. It can be seen from the output indexes that these three provincial PGEs focus on the final investment benefits rather than efficiency when investing. Because this stage does not consider the impact of external environment variables, random errors and other factors, it cannot reflect the real investment efficiency of these enterprises. In the second stage, therefore, after eliminating external environmental variables, random errors, and management inefficiencies through SFA regression model, the original investment of the PGEs will be adjusted accordingly.

### 5.2. The Second Stage: The Analysis of Impact of Environment Variables on Investment Efficiency

At this stage, the SFA model is used to perform regression analysis on the input slack variables (interpreted variables) separated in the first stage including power grid infrastructure investment, size of employees, 35 kV and above substation capacity. Three external environmental variables, GDP per capita, the proportion of industrial electricity consumption, and electrification rate, are used as explanatory variables. The SFA regression results are shown in Table 2:

It can be concluded from Table 1 that both the log likelihood function and the LR test are notable at 1%, indicating that the regression results are valid. The value of $\gamma$ of the three input variables are close to 1, indicating that the main factor contributing to the investment inefficiency of DMUs is management inefficiency.

According to Table 1, the coefficient of each input variable is notable at 1%, indicating that external environment variables have a strong impact on the investment efficiency of provincial PGEs. Therefore, to accurately assess the investment efficiency of these provincial PGEs, the uncontrollable external environment variables must be eliminated. Moreover, the coefficients of these slack variables indicate the relation between the input variables and the external environment variables. If the coefficient is positive, the value of the interpreted variable increases when the external environment variable increases, resulting in a decrease in investment efficiency. Otherwise, when the coefficient is less than zero, the value of the slack variable decreases as the external environment variable increases, resulting in an improvement in investment efficiency.

The effect of each external environment variable on the input variable is discussed as follows:

(1)  GDP per capita. Coefficient of GDP per capita is greater than zero and less than 1, indicating that the increase in GDP per capita is positively correlated with the slack value of input variables such as grid infrastructure investment, 35 kV and above substation capacity, and the size of employees. With the improvement of the living standards of residents in various provinces and regions, the investment efficiency of provincial PGEs will be reduced. Therefore, it can be concluded that the economic development of each province will increase residents' demand for electricity, and thus increase the investment of PGEs, but it will have a negative impact on investment efficiency of these enterprises.

(2)  The proportion of industrial electricity consumption. The proportion of industrial electricity consumption is greater than zero, indicating that the increase in the proportion of industrial electricity consumption in the total electricity consumption will promote the increase of slack variables such as power grid infrastructure investment, 35 kV and above substation capacity and size of employees. However, the impact of industrial electricity consumption is much greater than that of GDP per capita. This relation shows that the improvement of the industrialization level of the provinces and regions will promote the investment of PGEs, but hinder their investment efficiency.

(3)  Electrification rate. The effect of the electrification rate is exactly the opposite of that of the first two factors because the coefficients of the electrification rate are all less than zero. When the level of social electrification increases, the slack values of the above three input variables will be lower, resulting in an increase in investment efficiency of PGEs. The increase of proportion of energy consumption in the final energy consumption of various provinces and regions results in the increase of residents' expenditure on power consumption and in the sales of PGEs, thereby improving the investment efficiency of PGEs.

Overall, the environmental differences and errors result in differences in investment efficiency in different provincial PGEs. In order to ensure the accuracy of the investment efficiency evaluation of PGEs and get their real investment efficiency, it is necessary to eliminate the influence of uncontrollable variables such as external environment variables on the evaluation results.

*5.3. The Third Stage: The Real Investment Efficiency*

The operation process of this stage is consistent with that of the first stage. It only uses the adjusted value of the second stage to replace the original input value for BCC model analysis, and finally obtains more actual investment efficiency. For further comparison, the efficiency values of the first stage and the third stage are put together in Table 3.

After excluding the external environment variables and random errors, the average of the real investment efficiency of 31 provincial PGEs is 0.788, the average value of pure technical efficiency is 0.906, and the average scale efficiency is 0.865, indicating that the investment efficiency of Chinese provincial PGEs is low, which is mainly caused by scale efficiency. Comparing the eastern, central, and western region, the region with highest investment efficiency is eastern region, with an average of 0.769, followed by the western region, while the central region has the lowest average investment

efficiency. Tianjin, Liaoning, Jiangsu, and Fujian have relatively higher investment efficiency, while Shandong and Shanghai exhibit lower investment efficiency.

**Table 3.** The investment results of the first stage and the third stage.

| Regions | DMUs (Province) | The Overall Efficiency | | Pure Technical Efficiency | | Scale Efficiency | | Returns to Scale | |
|---|---|---|---|---|---|---|---|---|---|
| | | I | III | I | III | I | III | I | III |
| Eastern | Beijing | 1 | 0.764 | 1 | 1 | 1 | 0.764 | - | drs |
| | Tianjin | 1 | 1 | 1 | 1 | 1 | 1 | - | - |
| | Hebei | 0.892 | 0.883 | 1 | 1 | 0.892 | 0.883 | drs | drs |
| | Shandong | 0.438 | 0.463 | 0.6 | 0.609 | 0.731 | 0.761 | drs | drs |
| | Liaoning | 1 | 1 | 1 | 1 | 1 | 1 | - | - |
| | Jiangsu | 1 | 1 | 1 | 1 | 1 | 1 | - | - |
| | Shanghai | 0.473 | 0.474 | 0.635 | 1 | 0.744 | 0.474 | drs | drs |
| | Zhejiang | 0.63 | 0.665 | 0.958 | 0.926 | 0.658 | 0.719 | drs | drs |
| | Fujian | 0.967 | 1 | 1 | 1 | 0.967 | 1 | drs | - |
| | Hainan | 1 | 0.577 | 1 | 0.695 | 1 | 0.831 | - | drs |
| | Guangdong | 0.58 | 0.64 | 1 | 1 | 0.58 | 0.64 | drs | drs |
| | mean | 0.816 | 0.769 | 0.927 | 0.930 | 0.870 | 0.825 | | |
| Central | Shanxi | 0.908 | 0.702 | 1 | 0.891 | 0.908 | 0.789 | drs | drs |
| | Anhui | 0.865 | 0.658 | 1 | 1 | 0.865 | 0.658 | drs | drs |
| | Hubei | 0.574 | 0.695 | 0.761 | 0.907 | 0.754 | 0.767 | drs | drs |
| | Hunan | 0.972 | 0.731 | 1 | 0.752 | 0.972 | 0.973 | drs | drs |
| | Henan | 0.445 | 0.738 | 0.605 | 0.83 | 0.735 | 0.889 | drs | drs |
| | Jiangxi | 0.725 | 0.67 | 1 | 0.948 | 0.725 | 0.707 | drs | drs |
| | Chongqing | 0.966 | 0.897 | 0.992 | 0.899 | 0.974 | 0.999 | drs | drs |
| | Jilin | 0.971 | 0.682 | 0.971 | 0.682 | 1 | 1 | - | - |
| | Heilongjiang | 0.782 | 0.634 | 1 | 1 | 0.782 | 0.634 | drs | drs |
| | mean | 0.800 | 0.712 | 0.925 | 0.879 | 0.857 | 0.824 | | |
| Western | Inner Mongolia | 1 | 1 | 1 | 1 | 1 | 1 | - | - |
| | Sichuan | 0.515 | 0.537 | 0.634 | 0.689 | 0.813 | 0.778 | drs | drs |
| | Shaanxi | 0.792 | 0.614 | 1 | 1 | 0.792 | 0.614 | drs | drs |
| | Gansu | 0.926 | 0.781 | 1 | 0.922 | 0.926 | 0.848 | drs | drs |
| | Qinghai | 1 | 0.852 | 1 | 0.88 | 1 | 0.968 | - | drs |
| | Ningxia | 1 | 1 | 1 | 1 | 1 | 1 | - | - |
| | Xinjiang | 0.88 | 0.852 | 0.899 | 0.852 | 0.979 | 1 | drs | - |
| | Tibet | 1 | 1 | 1 | 1 | 1 | 1 | - | - |
| | Guizhou | 1 | 0.815 | 1 | 0.816 | 1 | 0.999 | - | irs |
| | Yunnan | 0.942 | 1 | 0.986 | 1 | 0.955 | 1 | drs | - |
| | Guangxi | 0.916 | 0.779 | 0.927 | 0.78 | 0.988 | 0.999 | drs | irs |
| | mean | 0.906 | 0.739 | 0.950 | 0.904 | 0.950 | 0.928 | | |
| | Total mean | 0.844 | 0.788 | 0.934 | 0.906 | 0.895 | 0.865 | | |

Note: drs implies the return to scale is decreasing; irs indicates the return to scale is increasing; "-" represents the return to scale is constant.

## 5.4. Comprehensive Analysis

The three-stage DEA model calculated the real investment efficiency value of China's provincial PGEs. According to Table 2, this part mainly discusses the impact of external environment variables and random errors on investment efficiency of PGEs.

From the comparison between the first stage and the third stage, variables such as external environment variables and random errors have a significant impact on the investment efficiency of PGEs. According to Table 3, the overall efficiency, technical efficiency, and scale efficiency of provincial PGEs decrease 5.6%, 2.8%, and 3% respectively. It suggests that in the first stage, the investment efficiency of provincial PGEs is overestimated.

Adjusted input efficiency values in the eastern, central and western regions suggest that the overall investment efficiency of provincial PGEs in the eastern region has decreased by 4.7%, the pure technical efficiency has increased slightly, and the scale efficiency has decreased by 4.5%, resulting in a decline in the investment efficiency of eastern region PGEs. The average investment efficiency of provincial PGEs in the central region decreased by 8.8%, and the pure technical efficiency and

scale efficiency also decreased. After adjusting the input value, the average investment efficiency of provincial PGEs in the western region dropped by 16.7%, and the external environment variables had a greater impact on its investment efficiency. Since the economic development, the process of industrialization, the geographical location and other factors in the western region lag behind those of the other two regions, there is a gap between the western region and the eastern/central regions in investment efficiency.

In terms of provincial differences, the investment efficiency of Tianjin, Liaoning, Jiangsu, and Inner Mongolia (the efficiency values of both the first stage and third stage are equal to 1) remains unchanged, and these enterprises have the most favorable investment efficiency. Excluding external environment variables, the investment efficiency of PGEs in Fujian and Yunnan has significantly improved with the highest investment efficiency, indicating that external environment variables of these provinces will reduce the investment efficiency of PGEs. Therefore, the uncontrollable variables must be excluded. In the first stage of assessments, the efficiencies of four provinces including Beijing, Hainan, Qinghai, and Guizhou are at the forefront, but after adjustment, it is found that in the first stage, the investment efficiency of these four provinces was seriously overestimated.

## 6. Discussion

Through three-stage DEA model, this paper develops an investment efficiency index system of the PGEs under the NEMR. The marketized transaction power, electric energy replacement, power supply reliability rate and integrated line loss rate are selected as output indexes. Grid infrastructure investment, 35 kV and above substation capacity and the size of employees are selected as input indexes. GDP per capita, industrial electricity consumption and electrification rate are selected as external environment variables. Through BCC model and SFA regression analysis, this paper calculated the real investment efficiency level of China's provincial PGEs under the NEMR.

From the comparison of Figures 3 and 4, it can be seen that the external environment variables and random errors have significant influence on China's provincial PGEs. However, the traditional DEA model ignores the impact of external environment variables and the random error terms on the investment efficiency of PGEs, resulting in an inaccurate evaluation of investment efficiency.

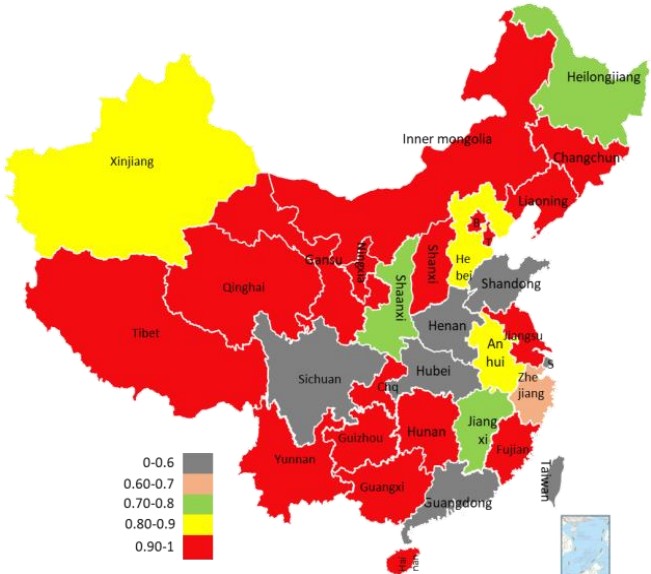

**Figure 3.** Investment efficiency of the first stage of China's provincial PGEs.

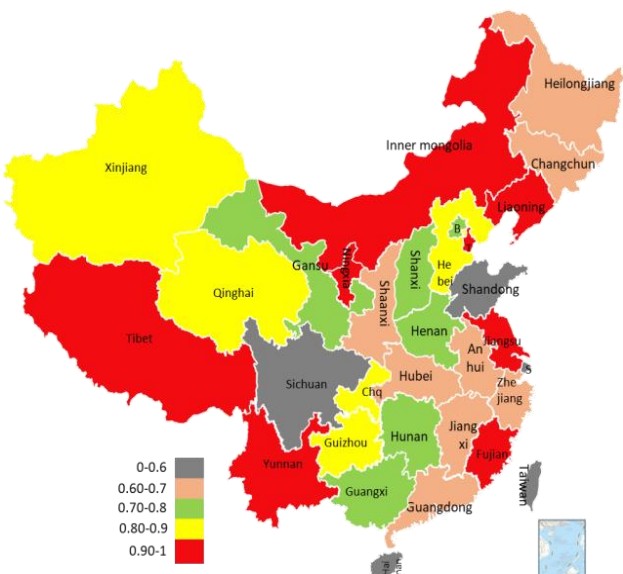

**Figure 4.** Investment efficiency of the third stage of China's provincial PGEs.

The specific discussion is as follows:

(1) In the first stage, the average investment efficiency of China's provincial PGEs is 0.844, the eastern region is 0.816, the central region is 0.8, and the western region is 0.906. Investment efficiencies of provincial PGEs in eastern and central regions are lower than the national average investment efficiency, mainly because the investment efficiencies of Shandong, Shanghai, and Henan are lower than 50%. But the investment efficiencies of 21 provinces are higher than the national average investment efficiency.

(2) In the second stage, through the SFA regression analysis, the regression coefficients are all notable at 1%, indicating that the external environment variables will have a significant impact on the investment efficiency of provincial PGEs. Regression coefficients of two external environment variables named GDP per capita and the proportion of industrial electricity, and three input slack variables are greater than zero, which is not conducive to the improvement of investment efficiency of PGEs. Among these indexes, the proportion of industrial electricity is more important. Moreover, the regression coefficients of electrification rate, power grid infrastructure investment, 35 kV and above substation capacity, and the size of employees are all negative, which will reduce the slack value of input variables, thus improving the investment efficiency level of PGEs.

(3) From the calculation of the third stage, it is found that the investment efficiency of most PGEs in the first stage is generally overestimated. The western region is the most obvious area affected by external environment factors, followed by the central region, and the eastern region is the least affected area. From Figures 3 and 4, it can be concluded that after excluding external environment variables, the provincial PGEs with investment efficiencies of greater than 90% and less than 60% are significantly reduced, and the investment efficiency of most PGEs is in 60%–90%.

(4) External environment variables and random errors have little impact on the investment efficiency of PGEs in Tianjin, Jiangsu, Liaoning, and Inner Mongolia. PGEs in Beijing, Hainan, Qinghai, and Guizhou are greatly affected by external environment variables. In the first stage, the investment efficiency of these provinces was overestimated. Due to the influence of external environment variables, Fujian and Yunnan only have low investment efficiencies. However, after eliminating the influence of external environment variables, their investment efficiencies increased significantly.

## 7. Conclusions

The implementation of the NEMR made higher requirements for the investment efficiency of China's provincial PGEs. Therefore, the investment effectiveness is the core criterion for investment plans, and the PGEs now attach great importance to the investment efficiency. Thus, this paper employed the three-stage DEA model to evaluate the investment efficiency of PGEs in 2017. Through the empirical research on PGEs investment efficiency, the following conclusions can be drawn:

(1) Based on the integrated investment efficiency, the average integrated efficiency of these provincial PGEs is 0.844, the average pure technical efficiency is 0.934, and the average scale efficiency is 0.895. And the integrated investment efficiencies of 10 provincial EGEs achieve high efficiency, while 21 provincial EGEs are inefficient.

(2) From the perspective of regional differences, the average integrated efficiency of provincial PGEs in the eastern, central, and western region is 0.816, 0.80, and 0.906 respectively.

(3) At the second stage, three external environment variables have considerable impacts on the investment efficiency (Except for Tianjin, Liaoning, Jiangsu, Inner Mongolia, Ningxia, and Tibet). And though the increase of GDP per capita and electricity consumption in industry are not conducive to improving investment efficiency, the advancement of electrification plays a positive role in its improvement.

(4) The real efficiency results, Tianjin, Liaoning, Jiangsu, and Fujian have relatively higher investment efficiency, while Henan, Shandong and Shanghai exhibit lower investment efficiency.

(5) The investment efficiency can be disintegrated into scale efficiency and pure energy efficiency. Meanwhile, the inefficient investments prevailed some provincial PGEs, which mainly caused by low scale efficiency.

This paper examined the investment efficiency of China's provincial PGEs in 2017. However, due to the relatively limited database, it does not consider more indices which have impact on the investment efficiency of PGEs under the NEMR, such as the number of new energy vehicles, R&D investment, agricultural network transformation and upgrading investment, and effective assets. These indices will have a certain degree of impact on the investment efficiency of PGEs, which can be the future research of this paper.

**Author Contributions:** Conceptualization, J.S. and N.R.; methodology, H.Z.; software, J.Z.; validation, N.R., J.Z. and H.Z.; formal analysis, B.S.; investigation, N.R.; resources and data curation, J.Z.; writing—original draft preparation, N.R.; writing—review and editing, H.Z.; visualization, B.S.; supervision, J.S.; project administration, J.S.; funding acquisition, J.S.

**Funding:** This research is sponsored by the science and technology project of State Grid Corporation of China named Theoretical, Model, and Application Research of Effective Incentives for Transmission and Distribution Tariff under New Electricity Reform.

**Acknowledgments:** This research is sponsored by the science and technology project of State Grid Corporation of China named Theoretical, Model, and Application Research of Effective Incentives for Transmission and Distribution Tariff under New Electricity Reform.

**Conflicts of Interest:** The authors declare no conflict of interest.

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
