# Peer review of "Evaluating the Investment Efficiency of China’s Provincial Power Grid Enterprises under New Electricity Market Reform: Empirical Evidence Based on Three-Stage DEA Model"

_energies, doi:10.3390/en12183524_

Round 1

Reviewer 1 Report

please find the attached file.

Author Response

Thank you for your valuable comments.

The question1, 2, 4, 5, 7, 8 we are revised in Corresponding location. The question3: China's eastern, central and western economies, development, and education levels are relatively large. The discussion in three regions is mainly to eliminate the impact of regional differences on grid enterprise investment. It is more important to discuss China in three regions. On this basis, the local GDP per capita is selected as the external environmental variable to study the impact of regional economy on the investment efficiency of PEGs. If the regional difference is an external environmental variable, it is difficult to quantify, and it is impossible to calculate the impact on the investment efficiency of the PEGs.  The question 6: The integrated electric loss rate in the output variable is as small as possible, and contrary to other variables. Therefore, in order to consistent the output variable, take the reciprocal of the overall electric lose rate.

Reviewer 2 Report

The article examines the performance of power grid enterprises (PGE) in China under the new electricity market reform (NEMR) conditions. The study was based on 31 PGE excluding the Taiwan, Hong Kong, and Macao regions due to the lack of relevant data. The research was carried out using a three-stage DEA. The research was based on selected indicators regarding the size and quality of PGE on the input variables side, the quantity and reliability of electricity supplied on the output variables side and selected external environmental variables to remove stochastic noise and the environmental impact on the tested efficiency.

The selection of input and output variables and the BBC model used in stage I and III of the three-stage DEA allows the creation of a ranking of the effectiveness of individual PGEs in the field of electricity supply technology used. Also the selection of environmental variables is satisfactory.
The article may be of interest to the reader who wants to compare the effectiveness of PGE in individual regions in China, taking into account the territorial division into the east, center and west of the country, but without an in-depth analysis of changes in the effectiveness of provincial PGE in China.

General comments:
Data for 2017 allow the analysis of effectiveness in a selected research sample only in one year. It is not possible to analyze changes (improvement) in PGE's effectiveness since the introduction of NEMR in 2015. It should be noted that the three-stage DEA allows to take into account the dynamics of efficiency changes at the second stage of the method.
The effectiveness analysis can only be related to PGE in China - the question then arises whether there is a decision-making unit in this group that is a local or global leader, because the reader does not know what the DMU selection method was (selected or all)?

From the smaller remarks:

- in line 267, the authors refer to equation (7) and not to equation (5).
- figure 3 and 4 - understanding the results from a non-Chinese reader would be helped by demarcation with the lines of eastern, central and western provinces.

Author Response

感谢您宝贵的时间和意见。

The new round of electricity market reform in China has been in its fifth year since 2015, but the deployment of some provincial PGEs was completed at the end of 2016, such as the establishment of Power Market Trading Center. Because, the variables data in 2015 and 2016 is incomplete and it is impossible to effectively evaluate the investment efficiency of provincial grid enterprises. The data for 2018 has not yet been officially announced. Meanwhile, The 2017 data is relatively complete compared to other years. Therefore, this paper chooses 2017 as the investment efficiency of PGEs  under the new round of electricity market reform.

Because we are divided into East, Central and Western according to the level of local development and education, we think that the dividing line between the three regions does not make much sense.

Other issues we have modified in the appropriate location.

Round 2

Reviewer 1 Report

The manuscript seems to be properly revised, but there are still problems with citations.

For example, Dunnan, L et al. [4]Wang and Shi et al. [12].

The manuscript is acceptable, but I recommend to recheck citation formats.

Author Response

Thank you for your precious time and advice.

We have carefully revised the questions you have raised, please check them.